# Synthesis and Biological Evaluation of Novel 2-Amino-1,4-Naphthoquinone Amide-Oxime Derivatives as Potent IDO1/STAT3 Dual Inhibitors with Prospective Antitumor Effects

**DOI:** 10.3390/molecules28166135

**Published:** 2023-08-19

**Authors:** Ri-Zhen Huang, Qiao-Ling Liang, Xiao-Teng Jing, Ke Wang, Hui-Yong Zhang, Heng-Shan Wang, Xian-Li Ma, Jian-Hua Wei, Ye Zhang

**Affiliations:** 1Guangxi Key Laboratory of Drug Discovery and Optimization, Guangxi Engineering Research Center for Pharmaceutical Molecular Screening and Druggability Evaluation, School of Pharmacy, Guilin Medical University, Guilin 541199, Chinawk843265696@163.com (K.W.);; 2State Key Laboratory for the Chemistry and Molecular Engineering of Medicinal Resources, Collaborative Innovation Center for Guangxi Ethnic Medicine, School of Chemistry and Pharmaceutical Sciences of Guangxi Normal University, Guilin 541004, China; 3Department of Chemistry & Pharmaceutical Science, Guilin Normal College, Xinyi Road 15, Guilin 541001, China

**Keywords:** indoleamine 2,3-dioxygenase 1, signal transducer and activator of transcription 3, dual inhibitors, naphthoquinone-oxime derivatives, anticancer agents

## Abstract

Indoleamine-2,3-dioxygenase 1 (IDO1) and signal transducer and activator of transcription 3 (STAT3) have emerged as significant targets in the tumor microenvironment for cancer therapy. In this study, we synthesized three novel 2-amino-1,4-naphthoquinone amide-oxime derivatives and identified them as dual inhibitors of IDO1 and STAT3. The representative compound **NK3** demonstrated effective binding to IDO1 and exhibited good inhibitory activity (hIDO1 IC_50_ = 0.06 μM), leading to its selection for further investigation. The direct interactions between compound **NK3** and IDO1 and STAT3 proteins were confirmed through surface plasmon resonance analysis. A molecular docking study of compound **NK3** revealed key interactions between **NK3** and IDO1, with the naphthoquinone-oxime moiety coordinating with the heme iron. In the in vitro anticancer assay, compound **NK3** displayed potent antitumor activity against selected cancer cell lines and effectively suppressed nuclear translocation of STAT3. Moreover, in vivo assays conducted on CT26 tumor-bearing Balb/c mice and an athymic HepG2 xenograft model revealed that compound **NK3** exhibited potent antitumor activity with low toxicity relative to 1-methyl-L-tryptophan (1-MT) and doxorubicin (DOX). Overall, these findings provided evidence that the dual inhibitors of IDO1 and STAT3 may offer a promising avenue for the development of highly effective drug candidates for cancer therapy.

## 1. Introduction

In recent years, the field of cancer immunotherapy has garnered considerable attention due to the successful use of drugs that target immune checkpoints in clinical cancer therapy. These drugs include anti-PD1 (programmed cell death protein 1), anti-PDL1 (programmed death ligand 1), and anti-CTLA4 (cytotoxic T-lymphocyte-associated protein 4) [1,2]. Despite the advancements in immune checkpoint therapies, a significant number of patients with malignancies remain unresponsive to these treatments. In fact, only a limited proportion of patients derive significant benefits [3]. One possible explanation for the evasion of immune disruption by tumor cells lies in their capacity to develop diverse tactics to elude, hinder, or manipulate both innate and adaptive immunity alongside the immune checkpoint mechanisms prevailing in immunosuppressive tumor microenvironments [4]. There is a pressing need to discover novel cancer immunotherapy treatments that can enhance the immune system’s ability to recognize and attack cancer cells.

Indoleamine 2,3-dioxygenase 1 (IDO1) is an oxidoreductase that contains heme and serves as a catalyst for the degradation of tryptophan to kynurenine through the kynurenine pathway [5,6]. The ingestion of tryptophan via indoleamine 2,3-dioxygenase (IDO) results in the inhibition of effector T-cell reactions and amplifies immunosuppressive signals regulated by T regulatory cells [7]. IDO1-induced immune tolerance is a crucial mechanism for tumors to escape immune surveillance [8]. Research has demonstrated a correlation between the upregulation of IDO1 in both human tumors and antigen-presenting cells of the host and unfavorable prognosis, as well as parameters indicative of tumor progression [9,10,11]. In mouse tumor models, blocking IDO1 activity with small-molecule inhibitors has been successful in delaying metastasis development, impairing tumor outgrowth, and prolonging survival [12]. Additionally, combining IDO1 inhibitors with anticancer drugs has shown synergistic therapeutic effects to facilitate regression of tumors that are otherwise difficult to treat [13]. Therefore, IDO1 is considered an attractive target for cancer immunotherapy. 

Several potent small-molecule IDO1 inhibitors have been identified to effectively stimulate antitumor immunity. Some of these inhibitors, including indoximod, epacadostat, navoximod, BMS-986205, and PF-06840003, have been developed and tested in clinical trials (Figure 1) [14,15,16]. However, previous preclinical investigations have demonstrated that IDO1 inhibitors possess only moderate antitumor activity when administered as standalone agents [17]. Epacadostat, the most advanced compound in clinical development, showed promising anticancer activity in an early phase I/II study, but was disappointing in failing to achieve notable results in combination with pembrolizumab in a subsequent pivotal phase III study [18]. Despite the potential synergistic effects observed when utilizing IDO1 inhibitors in conjunction with other therapies, the efficacy of drug combination strategies is consistently hindered by intricate pharmacokinetics and drug–drug interactions [19]. Consequently, there has been considerable interest in the development of a singular therapeutic agent capable of simultaneously targeting two or more cooperative mechanisms to address this concern. 

An important limitation of checkpoint inhibitors is that these molecules do not target key molecular determinants of the tumor microenvironment, which is a major cause of immune deficiency, and focus only on modulating the immune synapse [20,21]. As a transcriptional regulator of several tumor-promoting factors, signal transducer and activator of transcription 3 (STAT3) is involved in multiple oncogenic signaling pathways and is constitutively activated in cancer and immune cells in the tumor microenvironment [22,23]. The regulation of numerous genes essential for tumor cell survival, proliferation, migration, and angiogenesis is attributed to STAT3. Targeting constitutively activated STAT3 in tumors has been demonstrated to directly induce tumor cell death and inhibit growth in vivo [24,25,26]. In addition, constitutively activated STAT3 inhibits the expression of immune activation mediators and enhances the production of immunosuppressive factors, thereby promoting STAT3 activity in specific subsets of immune cells [27]. This alteration in gene expression programs ultimately suppresses antitumor immune responses. It is worth noting that recent investigations have demonstrated the role of IL6 in regulating the transcriptional expression of STAT3 [28,29]. Therefore, the identification of dual IDO1 and STAT3 inhibitors potentially represents a new strategy for anticancer treatment by harnessing the benefits of both immunotherapy and STAT3 inhibition. 

In our previous work, a series of naphthoquinone aromatic amide-oxime derivatives co-targeting IDO1 and STAT3 were synthesized [30]. As part of our ongoing study to discover potential IDO1/STAT3 dual inhibitors, we synthesized three novel 2-amino-1,4-naphthoquinone amide-oxime derivatives (NK1-NK3) with the hydrophobic terminal amine group, which could be tolerated in the active site of IDO1 and STAT3 [16,25], and evaluated for their IDO1/STAT3 inhibitory activities. In addition, the IDO1/STAT3 inhibitory results were further comprehended with the aid of surface plasmon resonance (SPR), molecular docking studies, immunofluorescence, and in vivo antitumor efficacy evaluation, providing insights into their biological properties. Our experimental data unequivocally demonstrated that the most potent bifunctional inhibitor, NK3, directly bonds to IDO1 and STAT3, exhibiting potent in vivo antitumor efficacy in both CT26 tumor-bearing mice and athymic nude mice. Thus, NK3 presented an efficient anticancer immunochemotherapy agent. 

## 2. Results and Discussion

### 2.1. Chemistry

The synthesis of 2-amino-1,4-naphthoquinone amide-oxime derivatives is demonstrated in Figure 1. Initially, commercially available phthalic anhydride (**1**) was subjected to a reaction with l-phenylalanine (**2**) in the presence of acetic acid, resulting in the formation of compound **3**, as reported in the previous literature [31]. Subsequently, compound **3** was treated with oxalyl chloride to generate an acyl chloride, which was then reacted with primary amines to yield amide **4**. Finally, compound **4** was subjected to a reaction with hydrazine hydrate in the presence of ethanol at ambient temperature, leading to the formation of amide derivative **5**. Compound **5** underwent treatment with 1,4-Naphthoquinone (**6**) and triethylamine solution for a duration of 18 h at ambient temperature, resulting in the formation of intermedium **7**. Subsequently, derivative **7** was subjected to a reaction with hydroxylamine hydrochloride in ethanol at a temperature of 80 °C. Following a reaction time of 12 hours, the corresponding 2-amino-1,4-naphthoquinone amide-oxime derivatives (**NK1**–**NK3**) were successfully obtained. The structures of title compounds **NK1**–**NK3** were confirmed by ^1^H NMR, ^13^C NMR, and high-resolution mass spectrometry (HRMS) (Appendix A).

### 2.2. Inhibition of IDO1 Activity

The in vitro evaluation of the synthesized 2-amino-1,4-naphthoquinone amide-oxime derivatives **NK1**–**NK3** against human IDO1 was conducted using a standard enzymatic assay, as described in a previous report [32]. The potent 4-amino-*N*-(3-chloro-4-fluorophenyl)-*N*′-hydroxy-1,2,5-oxadiazole-3-carboximidamide (IDO5L), which has been extensively characterized as one of the earliest IDO inhibitors in the literature, served as a positive control [33]. The IC_50_ values obtained from the in vitro inhibition activities of **NK1**–**NK3** are summarized in Figure 2. Results showed that compound **NK3** exhibited the best inhibitory activity against IDO1, which was similar to that of IDO5L. The other two analogs (**NK1**, **NK2**) were also found as IDO1 inhibitors with, however, relatively lower inhibitory potency.

### 2.3. Direct Interactions between NK3 and IDO1

The most potent inhibitor, **NK3**, was selected for further analysis. A surface plasmon resonance (SPR)-based binding assay was conducted to validate the direct interaction between **NK3** and IDO1 protein using a Biacore T200 optical biosensor. This analytical method is valuable for determining the kinetic and thermodynamic parameters of ligand–protein complex formation and is widely employed in the study of enzyme–enzyme/inhibitor interactions [34]. The binding affinity between compound NK3 and IDO1 was assessed utilizing Biacore analysis software, accompanied by measurements of kinetic association and dissociation. As depicted in Figure 3, compound **NK3** efficiently interacted with the immobilized protein to induce a concentration-dependent response for association or dissociation, respectively. Compounds **NK3** and IDO1 had a K_D_ value of 0.16 µM, which suggested the strong binding affinity of compound **NK3** with IDO1. The binding assays confirmed the direct binding of compound **NK3** to IDO1 protein. 

### 2.4. Molecular Docking Study of NK3 and IDO1

With confirmation that compound **NK3** directly bound to IDO1 protein with potent IDO1 enzyme inhibitory activity, we performed molecular docking analyses to elucidate the possible binding mode of compound **NK3** with the IDO1 (PDB ID: 4PK5) [35]. As depicted in Figure 4, the naphthoquinone oxime group of compound NK3 exhibited coordination with heme iron through the oxygen atom via the oxygen atom in the catalytically active site, which is indispensable for IDO1 inhibitory potency. The quinone moiety of **NK3** is conveniently located deep in the hydrophobic pocket formed by Tyr126, Val130, Phe163, and Leu234, properly positioned to facilitate π-π interactions with amino acid residues Tyr126, Phe163, and Phe164. Furthermore, the NH of **NK3** formed one hydrogen bond with Gly262 and contributed to the inhibition of IDO1 activity. Notably, the phenyl ring of the phenylalanine fragment exhibited a greater degree of burial within an additional hydrophobic pocket, establishing interaction between Arg231, Phe226, and Ile354. In particular, the result showed that the phenyl ring formed π-π interactions with the side chain of Phe226. This may be required for the strong inhibition of IDO1 activity and is consistent with previous findings on the role of Phe226 [35]. Of note, a strong hydrogen bond was observed between the amide moiety of **NK3** and the 7-propionic acid group in the heme ring, with the propylamine side chain projected out of the active site toward the solvent. 

### 2.5. In Vivo Antitumor Efficacy of NK3 in CT26 Tumor-Bearing Mice

Given its promising in vitro enzymatic potencies, compound **NK3** was further evaluated for antitumor efficacy in vivo with a model by injecting immunocompetent BALB/c mice with CT26 murine colon carcinoma cells (Figure 5). CT26 cells, derived from the epithelial glands of BALB/C mice with colon cancer, are commonly employed as a model for investigating immunotherapies and studying host immune responses [36]. Compound **NK3** and the positive control compound 1-MT and IDO5L were administered by intraperitoneal injection every 3 days for 21 consecutive days. The final tumor tissue size shown in Figure 5A clearly revealed the good tumor growth-inhibitory potency of **NK3** against CT26 tumor-bearing mice. Dose-dependent growth-inhibitory potency of **NK3** was observed (Figure 5B). After a duration of 21 days of treatment, it was observed that the tumor weight experienced a reduction of 42.8% when administered at a dosage of 50 mg/kg of compound **NK3**, and the tumor weight reduction rate reached 65.8% when the dose of **NK3** increased to 100 mg/kg. Interestingly, when treated with the positive control 1-MT at a dose of 100 mg/kg, the reduction in tumor weight (45.6%) was significantly lower than that of treatment with 100 mg/kg of **NK3**. Compound **NK3** dosed at 100 mg/kg exhibited comparable growth-inhibitory potency on tumors with IDO5L (65.8% for **NK3** vs. 70.9% for IDO5L). The relative tumor volume growth rate (T/C) of **NK3** was found to be 56.00% and 29.07% at dosages of 50 and 100 mg/kg, respectively, indicating a slightly weaker effect compared to IDO5L (26.05% at 100 mg/kg) (Figure 5C). Notably, there was no significant change in body weight of mice treated with both **NK3** and the positive control compounds compared to mice treated with the vehicle treatment, confirming the safety of **NK3** (Figure 5D). In addition, the pathological images of the heart, liver, spleen, lung, kidney and other important tissues of mice treated with compound **NK3** showed no obvious morphological and pathological changes (Figure 6). The above results indicated that **NK3** demonstrated potent antitumor efficacy and exhibited low toxicity in mice bearing CT26 tumors.

### 2.6. Cytotoxicity Assay

The cytotoxic activity of these three compounds was evaluated against HepG2, Hct-116, and SKOV3 cancer cells by MTT assay, using doxorubicin (DOX) as the positive control. As shown in Table 1, DOX was active in the low micromolar range, whereas IDO5L was inactive against the four solid tumor cell lines because IDO1 inhibitors do not destroy tumor cells directly. The data presented in Table 1 demonstrate the potent cytotoxicity of these three compounds across all examined cell lines. Particularly, **NK3** was identified as the most potent depending on the cell line, with IC_50_ values of 0.16 ± 0.04, 0.18 ± 0.04, and 0.48 ± 0.23 μM against HepG2, Hct-116, and SKOV3, respectively. 

### 2.7. Antitumor Potency of NK3 in Nude Mice In Vivo 

In order to ascertain the viability of IDO1 as an alternative target for compound **NK3**, a xenograft model was established by inoculating nude mice with human liver cancer HepG2 cells, subsequent to two injections of two doses of **NK3** administered at three-day intervals [38]. As shown in Figure 7A,B, intravenous injection of **NK3** at 10 mg/kg or 20 mg/kg every 3 days for 21 consecutive days significantly inhibited tumor growth (61.1% and 64.4%), which indicated that **NK3** treatment displayed potent antitumor efficacy. The antitumor effect of **NK3** was also reflected in a delayed increase in xenograft volume (Figure 7C). Notably, **NK3** was well tolerated and did not cause significant weight loss compared to DOX (Figure 7D). Taken together, these results demonstrated **NK3** significantly suppressed HepG2 xenografts’ tumor growth in a dose-dependent manner. These results also indicated that the antitumor efficacy of **NK3** did not require T-cell involvement, as IDO1 inhibitors did not destroy tumor cells directly [12,39]. However, our study successfully demonstrated the potential inhibitory effect of NK3 on IDO1 based on the aforementioned findings, which was found to be dependent on the functional integrity of T-cells in suppressing tumor growth in wild-type BALB/c mice. These findings suggested that other targets may be involved in **NK3**-induced tumor clearance.

### 2.8. Compound NK3 Directly Bind with STAT3

It has been reported that naphthoquinone-based compounds could serve as direct small-molecule inhibitors of IDO1 and STAT3 [38,40]. Recent research has provided evidence indicating that STAT3 plays a crucial role in regulating immune suppression induced by tumors within the tumor microenvironment [22]. Given that the STAT3 pathway plays a crucial role in various cancer-related processes, such as tumor cell proliferation, survival, angiogenesis, and invasion, it is plausible to suggest a direct association between conventional tumorigenesis and immunosuppression mediated by STAT3. Furthermore, the transcriptional expression of IDO and PD-L1 in human cancer is mediated by STAT3 [41,42]. In our recently published work, we identified naphthoquinone aromatic amide-oxime derivatives as dual inhibitors of IDO1 and STAT3, which exerted both immuno-modulatory and conventional chemotherapy effects [30]. The remarkable synergistic antitumor effects observed with **NK3** led us to propose the possibility of STAT3 being a potential alternative target of **NK3**.

In order to further investigate the direct interaction between **NK3** and STAT3, we conducted an SPR binding assay. The result revealed a strong dose-dependent binding mode between these two molecules, as demonstrated by the resonance curves in the BIAcore sensorgram (Figure 8). The measured *K*_D_ values, obtained through BIAcore evaluation software, indicate a high binding affinity of 0.17 µM. Collectively, these results certified the hypothesis that **NK3** specifically bound STAT3 protein with high affinity, thus suggesting that STAT3 was also the potential target of **NK3**. 

### 2.9. Compound NK3 Inhibited Nuclear Translocation of STAT3 

The central role of nuclear translocation in the functioning of transcription factors is evident. The activation of STAT3 transcription is reliant on its nuclear translocation and subsequent binding to target DNA [43], so we further investigated the inhibitory effect of **NK3** on preventing STAT3 from translocating to the nucleus by immunofluorescence staining. To this end, HepG2 cells were treated with 2 μM **NK3** for 6 h, and then IL-6 was added to stimulate STAT3 translocation for 30 min. As shown in Figure 9, strong nuclear fluorescence was observed in IL-6-treated HepG2 cells, indicating p-STAT3 nuclear translocation. STAT3 nuclear translocation was significantly inhibited in **NK3**-treated cells compared with IL-6-stimulated cells, suggesting that **NK3** suppressed STAT3 nuclear translocation. 

## 3. Experimental Section

### 3.1. General Information

All chemicals and solvents were procured from commercial sources and utilized without additional purification unless otherwise specified. Melting points were measured using the WRS-IA apparatus without any adjustments. NMR spectra were recorded in DMSO-*d*_6_ or CD_3_OD on AVANCE AV 400 (Bruker, Switzerland) with TMS as internal standard. HRMS were measured in FTMS EI or ESI mode, and the mass analyzer of the HRMS was TOF. Flash column chromatography was performed on silica gel (200−300 mesh). 

### 3.2. General Procedure for the Preparation of Compounds ***NK1***–***NK3***

Compound **3** (1 mmol) was dissolved in dry CH_2_Cl_2_ (15 mL) and stirred in an ice bath. After it was completely dissolved, oxaloyl chloride (1.5 mmol) was added. After stirring at room temperature for 6 h, the solvent and excess oxaloyl chloride were evaporated under reduced pressure. The acylated product was dissolved with dichloromethane, transferred to a constant pressure burette, drip added into a round-bottom flask containing primary amines (1 mmol) and triethylamine (0.5 mmol) under an ice bath condition, stirred at room temperature for 0.5 h, and rotated under pressure to obtain compound **4**. Then, to obtain the crude product, the mixture underwent evaporation under reduced pressure, followed by further purification through chromatography on silica gel eluted with petroleum ether/ethylacetate (*V*:*V* = 6:1), resulting in the formation of compound **4**. Compound **4** (1 mmol) was dissolved into anhydrous ethanol, then hydrazine hydrate (3 mmol) was added at room temperature following stirring. The mixture was stirred at room temperature for a duration of 8 h, after which the solvent was evaporated under reduced pressure. The crude product was then subjected to purification through silica gel chromatography eluted with petroleum ether/ethyl acetate (*V*:*V* = 3:1) to acquire compound **5**. A combination of Compounds **5** (2 mmol) and 1,4-naphthoquinone (3 mmol) was introduced into the mixture of triethylamine, DMF, and distilled water, which was stirred at room temperature for 18 h. The reaction process was detected by TLC. After the reaction was completed, its pH was adjusted to 3~4 with 1 mol/L hydrochloric acid, and then the aqueous layer was extracted with water and dichloromethane three times (30 mL × 3). Following drying with anhydrous sodium sulfate, the solvent was evaporated under reduced pressure to yield the residue, which was subjected to chromatography on a silica gel column (using a mixture of petroleum ether and ethyl acetate in a ratio of 4:1) to isolate compounds **7**. Compound **7** (1 mmol) and hydroxylamine hydrochloride (1 mmol) were added to absolute ethanol (25 mL). The reaction mixture was refluxed at 80 °C for 12 h and then diluted with water after evaporating under reduced pressure. The aqueous layer was extracted with methylene chloride (30 mL × 3), dried over anhydrous sodium sulfate, and evaporated to give the residue. The residue was chromatographed on silica gel column (light petroleum/ethyl acetate, *V*:*V* = 4:1) to obtain the title compounds **NK1**–**NK3**. The structures of title compounds were verified by ^1^H NMR, ^13^C NMR, and HR-MS. 

2-((1,4-dioxo-1,4-dihydronaphthalen-2-yl)amino)-*N*-(2-ethylhexyl)-3-phenylpropanamide (**7a**):



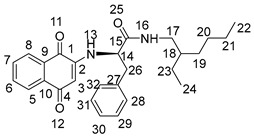



Yield: 70.3%. Yellow solid, m.p. 149.5~150.8 °C. [α${]}_{D}^{20}$ = −54 (*c* 0.1, AcOEt). ^1^H NMR (400 MHz, CD_3_OD) δ 7.97–7.85 (m, 2H, H-6,7), 7.68 (m, *J* = 7.6, 1.2 Hz, 1H, H-8), 7.60 (m, *J* = 7.5, 1.2 Hz, 1H, H-5), 7.28 (d, *J* = 4.4 Hz, 4H, H-28, 29, 31, 32), 7.24–7.19 (m, 1H, H-30), 5.64 (s, 1H, H-3), 4.25 (dt, *J* = 7.1, 3.7 Hz, 1H, H-14), 3.26–3.10 (m, 3H, H-17, H-26), 3.04 (ddd, *J* = 13.5, 6.5, 1.4 Hz, 1H, H-26), 1.36 (m, *J* = 11.4, 5.8 Hz, 1H, H-18), 1.20 (m, *J* = 13.2, 6.4 Hz, 8H, H-19,20,21,23), 0.88–0.77 (m, 6H, H-22,24). ^13^C NMR (101 MHz, CD_3_OD) δ 184.81 (s, C-1), 181.90 (s, C-4), 172.23 (s, C-15), 149.03 (s, C-2), 137.58 (s, C-27), 135.79 (s, C-7), 134.33 (s, C-6), 133.55 (s, C-9), 131.74 (s, C-10), 130.37 (s, C-29,31), 129.70 (s, C-28,32), 128.16 (s, C-8), 127.31 (s, C-5), 126.81 (s, C-30), 102.17 (s, C-3), 58.89 (s, C-14), 43.61 (s, C-17), 40.42 (s, C-18), 39.34 (s, C-26), 31.85 (s, C-19), 29.88 (s, C-20), 25.00 (s, C-23), 24.00 (s, C-21), 14.42 (s, C-22), and 11.11 (s, C-24). HR-MS (*m*/*z*) (ESI): calcd for C_27_H_33_N_2_O_3_ [M + H]^+^: 433.2486; found: 433.2470.

2-((1,4-dioxo-1,4-dihydronaphthalen-2-yl)amino)-*N*-phenethyl-3-phenylpropanamide (**7b**):



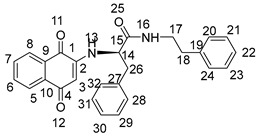



Yield: 60.2%. Yellow solid, m.p. 223.3~224.7 °C. [α${]}_{D}^{20}$ = −47 (*c* 0.1, AcOEt). ^1^H NMR (400 MHz, CD_3_OD) *δ* 7.92 (m, *J* = 8.0, 0.9 Hz, 2H, H-6,7), 7.69 (m, *J* = 7.6, 1.3 Hz, 1H, H-8), 7.60 (m, *J* = 7.5, 1.3 Hz, 1H,H-5), 7.30–7.21 (m, 5H,H-20,21,22,23,24), 7.21–7.13 (m, 3H,H-16,29,31,), 7.11 (t, *J* = 4.1 Hz, 2H,H-28,32), 7.09–7.04 (m, 1H,H-30), 5.55 (s, 1H,H-3), 4.16 (dd, *J* = 7.5, 6.4 Hz, 1H,H-14), 3.44 (dd, *J* = 13.9, 6.6 Hz, 1H,H-26), 3.37 (dd, *J* = 13.8, 6.7 Hz, 1H,H-26), 3.17 (dd, *J* = 13.7, 6.1 Hz, 1H,H-17), 3.08 (dd, *J* = 13.7, 7.8 Hz, 1H,H-17), 2.71 (dd, *J* = 12.0, 4.9 Hz, 2H,H-18). ^13^C NMR (101 MHz, CD_3_OD) *δ* 184.88 (s, C-1), 181.88 (s, C-4), 172.18 (s, C-15), 149.04 (s, C-2), 140.12 (s, C-19), 137.61 (s, C-27), 135.79 (s, C-7), 134.36 (s, C-6), 133.56 (s, C-9), 131.77 (s, C-10), 130.36 (s, C-21, 23), 129.77 (s, C-29, 31), 129.70 (s, C-20, 24), 129.46 (s, C-28, 32), 128.17 (s, C-8), 127.34 (s, C-5), 127.30 (s, C-22), 126.81 (s, C-30), 102.15 (s, C-3), 58.90 (s, C-14), 41.97 (s, C-17), 39.24 (s, C-26), and 36.27 (s, C-18). HR-MS (*m*/*z*) (ESI): calcd for C_27_H_25_N_2_O_3_ [M + H]^+^: 425.1860; found: 425.1842.

2-((1,4-dioxo-1,4-dihydronaphthalen-2-yl)amino)-3-phenyl-*N*-propylpropanamide (**7c**):



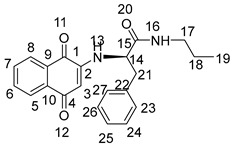



Yield: 55.4%. Yellow solid, m.p. 194.8~196.2 °C. [α${]}_{D}^{20}$ = −38 (*c* 0.1, AcOEt). ^1^H NMR (400 MHz, DMSO-*d*_6_) *δ* 8.16 (t, *J* = 5.6 Hz, 1H, H-13), 7.96 (dd, *J* = 7.6, 0.9 Hz, 1H, H-7), 7.90 (dd, *J* = 7.6, 1.0 Hz, 1H, H-6), 7.81 (m, *J* = 7.5, 1.3 Hz, 1H, H-8), 7.72 (m, *J* = 7.5, 1.3 Hz, 1H, H-5), 7.26 (d, *J* = 4.3 Hz, 4H, H-22, 23, 26, 27), 7.18 (dd, *J* = 8.6, 4.3 Hz, 1H, H-25), 7.07 (d, *J* = 8.2 Hz, 1H, H-16), 5.61 (s, 1H, H-3), 4.23 (dd, *J* = 14.1, 7.7 Hz, 1H, H-14), 3.14 (dd, *J* = 6.8, 2.9 Hz, 2H, H-17, 21), 3.02 (dd, *J* = 5.8, 3.3 Hz, 2H, H-17, 21), 1.44–1.29 (m, 2H, H-18), 0.78 (t, *J* = 7.4 Hz, 3H, H-19). ^13^C NMR (101 MHz, DMSO-*d*_6_) *δ* 181.74 (s, C-1), 181.05 (s, C-4), 169.30 (s, C-15), 147.49 (s, C-2), 137.21 (s, C-22), 134.94 (s, C-7), 132.75 (s, C-6), 132.46 (s, C-9), 130.22 (s, C-10), 129.24 (s, C-24, 26), 128.26 (s, C-23, 27), 126.59 (s, C-8), 125.98 (s, C-5), 125.38 (s, C-25), 100.80 (s, C-3), 56.96 (s, C-14), 40.47 (s, C-17), 37.23 (s, C-21), 22.19 (s, C-18), and 11.34 (s, C-19). HR-MS (*m*/*z*) (ESI): calcd for C_22_H_22_N_2_O_3_Na [M + Na]^+^: 385.1523; found: 385.1508.

(*Z*)-*N*-(2-ethylhexyl)-2-((4-(hydroxyimino)-1-oxo-1,4-dihydronaphthalen-2-yl)amino)-3-phenylpropanamide (**NK1**):



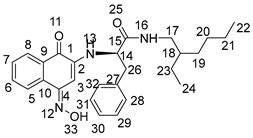



Yield: 50.5%. Yellow-green solid, m.p. 120.3~121.8 °C. [α${]}_{D}^{20}$ = −49 (*c* 0.1, AcOEt). ^1^H NMR (400 MHz, DMSO-*d*_6_) 12.30 (s, 1H, H-33), 8.16 (d, *J* = 7.6 Hz, 1H, H-13), 8.02 (m, *J* = 7.8, 3.8 Hz, 2H, H-5,8), 7.73–7.63 (m, 1H, H-6), 7.60–7.51 (m, 1H, H-7), 7.26 (d, *J* = 4.3 Hz, 4H, H-28, 29, 31, 32), 7.18 (m, *J* = 8.7, 4.3 Hz, 1H, H-30), 6.53 (d, *J* = 1.0 Hz, 1H, H-3), 6.16 (dd, *J* = 8.2, 1.6 Hz, 1H, H-16), 4.26–4.12 (m, 1H, H-14), 3.08 (d, *J* = 6.8 Hz, 2H, H-26), 3.04–3.00 (m, 1H, H-17), 2.94 (m, *J* = 18.8, 5.7 Hz, 1H, H-17), 1.30 (dd, *J* = 11.6, 5.8 Hz, 1H, H-18), 1.20–1.07 (m, 8H, H-19, 20, 21, 23), 0.76 (dd, *J* = 8.1, 5.7 Hz, 6H, H-22, 24). ^13^C NMR (101 MHz, DMSO-*d*_6_) δ 179.94 (s, C-1), 170.65 (s, C-15), 145.24 (s, C-4), 140.42 (s, C-2), 137.51 (s, C-27), 134.00 (s, C-10), 132.85 (s, C-9), 129.15 (s, C-29, 31), 128.90 (s, C-6), 128.47 (s, C-7), 128.29 (s, C-28, 32), 126.54 (s, C-8), 125.78 (s, C-5), 122.34 (s, C-30), 91.47 (s, C-3), 57.30 (s, C-14), 41.49 (s, C-17), 37.79 (s, C-18), 30.37 (s, C-26), 28.40 (s, C-19), 23.59 (s, C-20), 22.51 (s, C-23), 13.94 (s, C-21), 13.94 (s, C-22), and 10.77 (s, C-24). HR-MS (*m*/*z*) (ESI): calcd for C_27_H_34_N_3_O_3_ [M + H]^+^: 448.2595; found: 448.2576. Purity: 99.06%.

(*Z*)-2-((4-(hydroxyimino)-1-oxo-1,4-dihydronaphthalen-2-yl)amino)-*N*-phenethyl-3-phenylpropanamide (**NK2**):



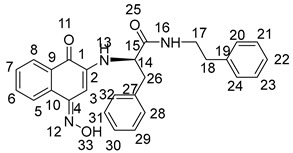



Yield: 49.5%. Yellow-green solid, m.p. 197.2~198.6 °C. [α${]}_{D}^{20}$ = −43 (*c* 0.1, AcOEt). ^1^H NMR (400 MHz, DMSO-*d*_6_) *δ* 12.32 (s, 1H, H-33), 8.23–8.15 (m, 2H, H-8, 13), 8.04 (dd, *J* = 7.9, 1.0 Hz, 1H, H-5), 7.70–7.66 (m, 1H, H-6), 7.59–7.55 (m, 1H, H-7), 7.28–7.22 (m, 4H, H-20, 21, 23, 24), 7.21–7.17 (m, 4H, H-28, 29, 31, 32), 7.14 (d, *J* = 6.7 Hz, 2H, H-22, 30), 6.50 (s, 1H, H-3), 6.16 (d, *J* = 8.1 Hz, 1H, H-16), 4.13 (dd, *J* = 13.7, 7.7 Hz, 1H, H-14), 3.35–3.25 (m, 2H, H-26), 3.04 (dd, *J* = 8.9, 5.5 Hz, 2H, H-17), 2.70–2.64 (m, 2H, H-18). ^13^C NMR (101 MHz, DMSO-*d*_6_) *δ* 180.30 (s, C-1), 170.96 (s, C-15), 145.73 (s, C-4), 140.85 (s, C-2), 139.75 (s, C-19), 137.88 (s, C-27), 134.39 (s, C-10), 133.32 (s, C-9), 129.59 (s, C-21, 23), 129.39 (s, C-6), 129.12 (s, C-29, 31), 128.91 (s, C-7), 128.74 (s, C-20, 24), 128.70 (s, C-29, 31), 126.98 (s, C-8), 126.51 (s, C-5), 126.23 (s, C-22), 122.78 (s, C-30), 91.71 (s, C-3), 57.71 (s, C-14), 39.32 (s, C-17), 38.08 (s, C-26), and 35.57 (s, C-18). HR-MS (*m*/*z*) (ESI): calcd for C_27_H_26_N_3_O_3_ [M + H]^+^: 440.1969; found: 440.1955. Purity: 96.97%.

(*Z*)-2-((4-(hydroxyimino)-1-oxo-1,4-dihydronaphthalen-2-yl)amino)-3-phenyl-*N*-propylpropanamide (**NK3**):



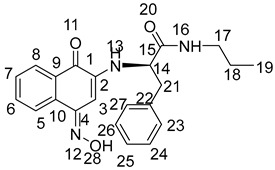



Yield: 30.2%. Yellow-green solid, m.p. 180.0~181.5 °C. [α${]}_{D}^{20}$= −34 (*c* 0.1, AcOEt). ^1^H NMR (400 MHz, DMSO-*d*_6_) *δ* 12.31 (s, 1H, H-28), 8.16 (t, *J* = 6.4 Hz, 2H, H-8, 13), 8.06–7.98 (m, 1H, H-5), 7.72–7.64 (m, 1H, H-6), 7.59–7.52 (m, 1H, H-7), 7.34–7.23 (m, 4H, 23, 24, 26, 27), 7.18 (ddd, *J* = 6.6, 5.0, 3.1 Hz, 1H, H-25), 6.51 (s, 1H, H-3), 6.16 (d, *J* = 8.0 Hz, 1H, H-16), 4.16 (q, *J* = 6.9 Hz, 1H, H-14), 3.09 (d, *J* = 6.7 Hz, 2H, H-21), 3.02 (dd, *J* = 12.7, 6.7 Hz, 2H, H-17), 1.43–1.31 (m, 2H, H-18), 0.78 (t, *J* = 7.4 Hz, 3H, H-19). ^13^C NMR (101 MHz, DMSO-*d*_6_) *δ* 179.91 (s, C-1), 170.33 (s, C-15), 145.27 (s, C-4), 140.40 (s, C-2), 137.42 (s, C-22), 133.94 (s, C-10), 132.89 (s, C-9), 129.19 (s, C-24, 26), 128.95 (s, C-6), 128.46 (s, C-7), 128.31 (s, C-23, 27), 126.57 (s, C-25), 125.81 (s, C-8), 122.35 (s, C-5), 91.26 (s, C-3), 57.16 (s, C-14), 40.43 (s, C-17), 37.73 (s, C-21), 22.26 (s, C-18), and 11.35 (s, C-19). HR-MS (*m*/*z*) (ESI): calcd for C_22_H_23_N_3_O_3_Na [M + Na]^+^: 400.1632; found: 400.1618. Purity: 98.40%.

### 3.3. Biological Assays

All adopted biological experimental procedures of enzymatic assays, SPR experiments, molecular docking, cell viability assay, immunofluorescence staining, and in vivo antitumor efficacy were performed according to our previous work [30,37].

## 4. Conclusions

In summary, we have presented three novel 2-amino-1,4-naphthoquinone amide-oxime derivatives as dual IDO1/STAT3 inhibitors. As a result, compound **NK3** exhibited the highest potency, with an IC_50_ value of 0.06 μM in the enzymatic assay against IDO1. Our findings indicated that the quinone oxime core of compound **NK3** played a crucial role in inhibiting IDO1, and the oxygen atom of the oxime group could serve as the iron binding group. Additionally, compound **NK3** showed strong binding affinities toward IDO1 and STAT3 through SPR analysis. Accordingly, the in vivo immunocompetent BALB/c mice and nude mice model indicated that compound **NK3** remarkably reduced tumor growth to a significant extent, signifying the multimodal action of anticancer and immuno-modulatory activity. Moreover, compound **NK3** significantly suppressed STAT3 nuclear translocation. Therefore, compound **NK3** could be a promising dual IDO1/STAT3 inhibitor for the development of novel targeted antitumor drugs.

## Data Availability

The data presented in this study are available on request from the authors.

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
