# Peer review of "Synthesis and Biological Evaluation of Novel 2-Amino-1,4-Naphthoquinone Amide-Oxime Derivatives as Potent IDO1/STAT3 Dual Inhibitors with Prospective Antitumor Effects"

_molecules, 2023, doi:10.3390/molecules28166135_

Round 1

Reviewer 1 Report

I can't judge the biological activity part.

As regards the synthetic part, the molecules are synthesized with a standard protocol of no synthetic interest. Only the yields of the intermediates 7 and of the adducts NK are reported. Instead, for each adduct NK, it would be necessary to report not only the total yields calculated from 3 but also the yields of the various intermediates 4,5,6. I have the impression that the total yields of NK are quite low and therefore this is not a particularly efficient synthetic protocol. Therefore, if the biological activity of NKs is confirmed, it will be necessary to find more efficient ways to obtain them.

However, if the goal was simply to obtain molecules for biological tests, it has been achieved.

The target molecules have been adequately characterized. 

A chiral center is present in intermediates 7 and in the target structures. I think the authors should evaluate the possibility of an enantioselective synthesis and check if both enantiomers have the same biological activity.

Author Response

(1) Question: As regards the synthetic part, the molecules are synthesized with a standard protocol of no synthetic interest. Only the yields of the intermediates 7 and of the adducts NK are reported. Instead, for each adduct NK, it would be necessary to report not only the total yields calculated from 3 but also the yields of the various intermediates 4,5,6.

Answer: We agree with the reviewer’ comment. The yields of the intermediates and title compounds have been provided in Scheme 1. Thank you for your comment.

(2) Question: I have the impression that the total yields of NK are quite low and therefore this is not a particularly efficient synthetic protocol. Therefore, if the biological activity of NKs is confirmed, it will be necessary to find more efficient ways to obtain them.

However, if the goal was simply to obtain molecules for biological tests, it has been achieved.

The target molecules have been adequately characterized.

Answer: This is a very good idea. More efficient ways to obtain these compounds are worthy of further investigation. In this manuscript, we just focus on the preparation of molecules for biological tests. Thank you for your comment.

(3) Question: A chiral center is present in intermediates 7 and in the target structures. I think the authors should evaluate the possibility of an enantioselective synthesis and check if both enantiomers have the same biological activity.

Answer: The target compounds were synthesized by L-amino acids. Thank you for your comment.

Reviewer 2 Report

I have many doubts:

1. „IC50 values obtained based on in vitro inhibition activities of NK1-NK3 were summarized in Fig. 1.”. It should be Fig. 2.

2. The results of in vitro and in vivo studies are not as satisfactory as the authors write. NK3 requires higher doses than dox to be satisfactory in vivo studies. It was not explained why these and not other drug concentrations were used. There is also no greater success in comparing the inhibitory effect of NK3 and IDO5L on IDO1. The IC50 of both drugs is similar.

3. I have serious doubts about the purity of the tested compounds. The signals in NMR are not assigned. There are additional signals that, in my opinion, indicate the lack of purity of the tested compounds. Also, signal integration is not adequate. What it comes from?

4. When working on unpurified compounds, you cannot conduct reliable research, much less test them on animals. This is unacceptable.

Minor editing of English language required.

Author Response

(1) Question: “IC50 values obtained based on in vitro inhibition activities of NK1-NK3 were summarized in Fig. 1.”. It should be Fig. 2.

Answer: We have carefully checked and revised this sentence. Thank you.

(2) Question: The results of in vitro and in vivo studies are not as satisfactory as the authors write. NK3 requires higher doses than dox to be satisfactory in vivo studies. It was not explained why these and not other drug concentrations were used. There is also no greater success in comparing the inhibitory effect of NK3 and IDO5L on IDO1. The IC50 of both drugs is similar.

Answer: We agree with the reviewer’ comment. We have carefully checked and revised this part. Thank you.

(3) Question: I have serious doubts about the purity of the tested compounds. The signals in NMR are not assigned. There are additional signals that, in my opinion, indicate the lack of purity of the tested compounds. Also, signal integration is not adequate. What it comes from?

Answer: We agree with the reviewer’ comment. We have assigned the NMR signals of all compounds. The purity of the tested compounds has been provided in manuscript and supplementary materials. Thanks for your comments.

(4) Question: When working on unpurified compounds, you cannot conduct reliable research, much less test them on animals. This is unacceptable.

Answer: We agree with the reviewer’ comment. We have analyzed the purity of the tested compounds by HPLC. The results showed that the purities were higher than 95%. Thanks for your comments. 

Reviewer 3 Report

This manuscript describes the synthesis and evaluation of three novel 2-amino-1,4-naphthoquinone amide-oxime derivatives (NK1-NK3) which showed IDO1/STAT3 inhibitory activities. The downside is that the structures of three derivatives in this work were very similar to those in their previous work (J. Med. Chem. 2020, 63, 1544), and only 3 compounds had been prepared for evaluation of activity.  However, compound NK3 showed good IC50 value against IDO1 and reduced tumor growth to a significant extent in both immunocompetent BALB/c mice and nude mice, it was promising dual IDO1/STAT3 inhibitor for the development of novel targeted antitumor drug. I recommend publication in Molecules.   

Minor corrections:
line 26: 'IC50=0.06', '50' should be subscript.
line 117-119 'To synthesize compound 4, compound 3 was treated with oxalyl chloride to produce phthalic imide acid chloride, which was substituted with primary amines to produce phthalic acid amide.' could be replaced by 'Compound 3 was treated with oxalyl chloride to produce an acyl chloride, which was reacted with primary amines to produce amide 4.'
line 120-125 need to be re-written.
line 312 'Compound 3 was synthesized according to the literature [31].' should be deleted. which has already been mentioned in chemistry part 2.1

Minor editing of English language required

Author Response

(1) Question: line 26: 'IC50=0.06', '50' should be subscript.

Answer: We have carefully checked and change 50 to be subscript. Thank you for your comments.

(2) Question: line 117-119 'To synthesize compound 4, compound 3 was treated with oxalyl chloride to produce phthalic imide acid chloride, which was substituted with primary amines to produce phthalic acid amide.' could be replaced by 'Compound 3 was treated with oxalyl chloride to produce an acyl chloride, which was reacted with primary amines to produce amide 4.'

Answer: We agree with the reviewer’ comment. We have replaced this sentence according to your comment. Thank you.

(3) Question: line 120-125 need to be re-written.

Answer: We have re-written these sentences. Thank you for your comments.

(4) Question: line 312 'Compound 3 was synthesized according to the literature [31].' should be deleted. which has already been mentioned in chemistry part 2.1

Answer: We agree with the reviewer’ comment. This sentence has been deleted. Thank you.

Reviewer 4 Report

The authors report the novel 2-amino-1,4-2-naphthoquinone amide-oxime derivatives. The target compounds have anticancer activity mediated by inhibition both IDO1 and STAT3. The manuscript is well-structured, interesting and contains new data. But this work must be improved according to the following comments:

1.         In Scheme 1 provide yields for all compounds.

2.         In Scheme 1, L.130: correct the “aromatic primary amines”.

3.         Specify why these primary amines were chosen for the synthesis of compound 4?

4.         L. 221, 227, 226, 272: correct the references.

5.         In “3.1. General information” only “DMSO-d6” is specified, while “MeOD” is also used in the experimental part. Also, instead of “MeOD” should be “CD3OD”. Please, provide more details about the instruments.

6.         Why does the melting point of NK1 have such a wide range ~10°C? How do you check the purity of compounds?

In general, this manuscript can be accepted for publication in this journal after minor revisions.

Author Response

(1) Question: In Scheme 1 provide yields for all compounds.

Answer: We agree with the reviewer’ comment. The yields of the intermediates and title compounds have been provided in Scheme 1. Thank you.

(2) Question: In Scheme 1, L.130: correct the “aromatic primary amines”.

Answer: We agree with the reviewer’ comment and sorry for our carelessness. We have carefully checked and change aromatic primary amines to primary amines. Thank you.

(3) Question: Specify why these primary amines were chosen for the synthesis of compound 4?

Answer: We agree with the reviewer’ comment. The reason has been provided in the manuscript. Thank you.

(4) Question: L. 221, 227, 226, 272: correct the references.

Answer: We agree with the reviewer’ comment. We have corrected these references. Thank you.

(5) Question: In “3.1. General information” only “DMSO-d6” is specified, while “MeOD” is also used in the experimental part. Also, instead of “MeOD” should be “CD3OD”. Please, provide more details about the instruments.

Answer: We agree with the reviewer’ comment. We have carefully checked and revised this part. The details about the instruments have been provided. Thank you.

(6) Question: Why does the melting point of NK1 have such a wide range ~10°C? How do you check the purity of compounds?

Answer: We agree with the reviewer’ comment. We have carefully checked and re-tested the melting point. The purities of compounds have also been provided. Thank you for your comments.

Round 2

Reviewer 1 Report

accept in  present form

Author Response

Thank you for your comment.

Reviewer 2 Report

Congratulations!

Author Response

Thank you for your comment.